# Partial Characterization, the Immune Modulation and Anticancer Activities of Sulfated Polysaccharides from Filamentous Microalgae *Tribonema* sp.

**DOI:** 10.3390/molecules24020322

**Published:** 2019-01-17

**Authors:** Xiaolin Chen, Lin Song, Hui Wang, Song Liu, Huahua Yu, Xueqin Wang, Rongfeng Li, Tianzhong Liu, Pengcheng Li

**Affiliations:** 1Key Laboratory of Experimental Marine Biology, Institute of Oceanology, Chinese Academy of Sciences, Qingdao 266071, China; chenxl@qdio.ac.cn (X.C.); sliu@qdio.ac.cn (S.L.); yuhuahua@qdio.ac.cn (H.Y.); xueqinwang@qdio.ac.cn (X.W.); rongfengli@qdio.ac.cn (R.L.); 2Laboratory for Marine Drugs and Bioproducts of Qingdao National Laboratory for Marine Science and Technology, Qingdao 266071, China; 3Center for Ocean Mega-Science, Chinese Academy of Sciences, Qingdao 266071, China; 4Shandong Provincial Key Laboratory of biochemical engineering, College of Marine Science and Biological Engineering, Qingdao University of Science and Technology, Qingdao 266042, China; lylinsong@hotmail.com; 5Qingdao Institute of Bioenergy and Bioprocess Technology, Chinese Academy of Sciences, Qingdao 266101, China; wanghui@qibebt.ac.cn (H.W.); liutz@qibebt.ac.cn (T.L.)

**Keywords:** characterization, immune-modulatory, anticancer, sulfated polysaccharide, *Tribonema* sp.

## Abstract

Recently, *Tribonema* sp., a kind of filamentous microalgae, has been studied for biofuel production due to its accumulation of triacylglycerols. However, the polysaccharides of *Tribonema* sp. and their biological activities have rarely been reported. In this paper, we extracted sulfated polysaccharides from *Tribonema* sp. (TSP), characterized their chemical composition and structure, and determined their immunostimulation and anticancer activities on RAW264.7 macrophage cells and HepG2 cells. The results showed that TSP is a sulfated polysaccharide with a M_w_ of 197 kDa. TSP is a heteropolysaccharide that is composed mainly of galactose. It showed significant immune-modulatory activity by stimulating macrophage cells, such as upregulating interleukin 6 (IL-6), interleukin 10 (IL-10), and tumor necrosis factor α (TNF-α). In addition, TSP also showed significant dose-dependent anticancer activity (with an inhibition rate of up to 66.8% at 250 µg/mL) on HepG2 cells as determined by the 3-(4,5-Dimethylthiazol-2-yl)-2,5-diphenyltetrazolium bromide (MTT) assay. The cycle analysis indicated that the anticancer activity of TSP is mainly the result of induced cell apoptosis rather than affecting the cell cycle and mitosis of HepG2 cells. These findings suggest that TSP might have potential as an anticancer resource, but further research is needed, especially in vivo experiments, to explore the anticancer mechanism of TSP.

## 1. Introduction

Due to limited fossil fuel resources, there is an increasing need for renewable energy sources, especially biofuels [1]. Microalgae are seen as a promising source of biomass for biofuels due to their advantageous features, such as their phototropic nature, lipid accumulation, high growth rate, and noncompetition with food crops for arable land [2,3]. During the past 30 years, many microalgae have been selected for their potential for biofuel production [4]. Recently, the filamentous algae *Tribonema* sp. has been reported for biofuel and palmitic and oleic acid production due to its distinctive properties of high lipid oil content, robustness to protozoa, and cost-effective harvesting [5,6,7]. In addition, *Tribonema* sp. contains carbohydrates mainly in the form of cellulose. However, there has been no report on its polysaccharides and their biological activity, especially sulfated polysaccharides. 

Polysaccharides from algae, including seaweeds and microalgae, have been reported to exhibit many biological activities, such as antiviral, antitumor, immunomodulation, hypoglycemic, anticoagulant, and antioxidant activities [8,9,10,11,12,13]. The polysaccharides from seaweeds have been widely studied because the massive culture and harvest are simple and the cost of seaweed has been lower than that of microalgae. For microalgae, because there are still some bottlenecks in their culture and utilization, relatively fewer researches were reported about their polysaccharides and associated activities compared with seaweeds polysaccharides. Despite this situation, there have been some references to the polysaccharides, including intracellular and exopolysaccharides, from *Spirulina platensis*, *Porphyridium cruentum*, *Arthrospira platensis*, and *Tetraselmis* species [14,15,16,17]. The polysaccharides from these microalgae have exhibited different biological activities, including antioxidant, immunomodulation, and anticancer activities. Nevertheless, various microalgae contain different chemical structures and biological activities. Therefore, it is not clear whether polysaccharides from *Tribonema* sp. Would have the same activities.

On the other hand, cancer is one of the most common diseases that threatens human life. There are more than 200 different types of cancers, with lung cancer, liver cancer, breast cancer, pancreatic cancer, and cervical carcinoma being the most common. These cancers are lethal because they can swiftly spread to other tissues and cause death [18]. Unfortunately, at present, cancer treatments, such as chemotherapy and radiotherapy, can cause many adverse reactions in patients. These drugs are toxic and affect not only cancer cells but also normal cells. Thus, it is important to find novel, effective, and nontoxic compounds from natural sources [19]. In recent years, there has been growing attention given toward searching for natural compounds in algae due to the specific structure and biological activity of algae. Furthermore, with advances in culture technology of microalgae, the price of polysaccharides from microalgae will decrease and compete with those from macroalgae and terrestrial plants [20]. Therefore, in order to further utilize *Tribonema* sp. in addition to being a resource for biofuel, we investigated it as a potential new anticancer resource. We extracted polysaccharides from *Tribonema* sp., partially characterized the chemical components and structure, and determined their immunomodulation and anticancer activities.

## 2. Results

### 2.1. Chemical Characterization of TSP

TSP was extracted and purified from *Tribonema* sp., with a yield of 1.5%. It was lower than that of *Spirulina platensis* algae (13.6%) [14], which might be due to a different algae series and extraction method. It was further characterized by molecular weight (M_w_), total sugar, sulfate content, and monosaccharide composition (Table 1).

According to Table 1, the total sugar and sulfate contents of TSP were 32.5% and 21.9%, respectively. The M_w_ of TSP was 197 kDa, which was not high, so it dissolved easily in water. The monosaccharide composition showed that galactose was the most prevalent monosaccharide, followed by xylose, and the other monosaccharides were present in minor quantities. These results indicated that TSP is a hybrid and sulfated polysaccharide.

To further characterize the chemical structure of TSP, its FTIR spectrum was determined (shown in Figure 1). The absorption peak at 3326 cm^−1^ corresponded to the O-H stretching vibration, and 2931 cm^−1^ corresponded to the absorption of the C-H stretching vibration. The absorption peaks at approximately 1600 cm^−1^ and 1411 cm^−1^ represented C=O asymmetric and symmetric stretching vibrations, respectively. The absorption peaks at 1245 cm^−1^ and 1075 cm^−1^ indicated S=O stretching vibration and C-O-H deformation vibration. The peak at 890 cm^−1^ was attributed to C-O-S symmetrical stretching vibration. These results further confirmed that TSP is a sulfated polysaccharide.

### 2.2. Immunological Activity of TSP

The effects of various concentrations of TSP on RAW264.7 macrophage viability and its cytokines are shown in Figure 2. The results indicate that different concentrations of TSP had variable effects on cell viability. Treatment with 25 µg/mL TSP significantly increased cell viability compared to the control. When the concentration of TSP increased, the cell viability decreased. The cell viability at 200 µg/mL of TSP was close to that of the control. Therefore, in subsequent cytokine determination experiments, TSP concentrations above 200 µg/mL were not used. As illustrated in Figure 2, the cytokines IL-6, IL-10, and TNF-α were significantly higher in treated cells than in control cells. IL-6 and IL-10 at 200 µg/mL TSP were even higher than LPS, which indicated that TSP could be a strong stimulant of RAW264.7 cells.

### 2.3. Anticancer Activity of TSP

#### 2.3.1. Evaluating the Inhibition of HepG2 Growth Activity In Vitro

Based on the immunological activity results, we further determined the anticancer effect of TSP on HepG2 cells. First, the antiproliferative effect of the polysaccharide on HepG2 cells was tested by the MTT assay. The results are shown in Figure 3. These results indicate that TSP had a dose-dependent antiproliferative effect on cancer cells. As the concentration increased, the inhibition rate increased. When the concentration was 250 µg/mL, the inhibition rate was up to 66.8%, which meant that TSP might be a good alternative anticancer resource. Therefore, we continued to study cell apoptosis and the cell cycle by flow cytometry with Guava^®^ easyCyte 6-2L (Millipore, Billerica, MA, USA).

#### 2.3.2. Cell Apoptosis Assessment Results

According to the results of flow cytometry, we examined the cell apoptosis rate by treating cells with different concentrations of TSP (Figure 4). As shown in Figure 4, when the concentration of TSP increased, the cell apoptosis rate also increased, although it did not increase at 150 µg/mL. When the concentration was above 200 µg/mL, the cell apoptosis rate was substantially increased. At 250 µg/mL, the rate was 38.65%, which indicates that TSP could obviously induce cell apoptosis. The results were consistent with those of the MTT assays. Next, we analyzed the cell cycle to further study the apoptosis induction mechanism. 

#### 2.3.3. Induction of Apoptosis According to Cell Cycle Analysis

In Figure 5, there was no significant influence of TSP on the cell cycle rate of HepG2 cells in contrast to the control regardless of concentration. These results might indicate that TSP exhibited anticancer effects mainly by inducing cell apoptosis rather than by affecting the mitosis of HepG2 cells.

## 3. Discussion

Polysaccharides are widely found in animals, plants, and microorganisms. As one of the components of living things, they have been discovered to be involved in various life processes, such as the information transmission and immune cell sensation, cell transformation, cell division, and cell regeneration. Among the polysaccharides, sulfated polysaccharides are bioactive macromolecules in which some hydroxyl groups are substituted with sulfate groups in the sugar residues. These anionic polysaccharides are ubiquitous in nature and exist in mammals, invertebrates, and flora. For algae, especially marine algae, sulfated polysaccharides are the major components of the cell wall [21,22,23]. Due to the growth of algae in special marine environments, the structure of sulfated polysaccharides is different from that found in terrestrial organisms. To date, these sulfated polysaccharides have been extensively studied because of their various biological activities, including their antitumor, anti-inflammation, and anticancer and immunomodulation activities [24,25,26,27,28]. Currently, immunomodulatory studies on sulfated polysaccharides from seaweeds have been performed mainly on polysaccharides from the brown algae *Undaria pinnatifida*, green algae *Caulerpa lentillifera*, and red seaweed *Hypnea spinella* [29,30,31]. However, the effect of microalgal sulfated polysaccharides on the human immune system has been less studied except in *Phaeodactylum tricornutum*, *Chlorella stigmatophora*, *Pavlova viridis*, *Thraustochytriidae*, and *Spirulina* [32,33,34,35]. Recently, *Tribonema* has been studied as a biofuel resource. However, there is no research on polysaccharides from *Tribonema*. To increase the value of this algae, in this paper, we extracted polysaccharides from *Tribonema* sp. and studied their immunomodulatory activity. The results showed that TSP is a sulfated polysaccharide, which is consistent with polysaccharides from other microalgae, but the sulfate concentration (21.9%) is higher than in *Pavlova viridis* (16%), *Phaeodactylum tricornutum* (<14%), and *Chlorella stigmatophora* (<10%). These results indicated that TSP has a high polyanionic character. For immunomodulatory activity, macrophages play a prominent role in defending against infection, boosting the stability of the organism, and immune monitoring. In this paper, TSP could increase RAW264.7 macrophage viability and stimulate cytokines such as IL-6, IL-10, and TNF-α. Similar to the polysaccharides of *Pavlova viridis*, it could also increase the proliferation of macrophages [33]. 

The occurrence, growth, and decline of cancer are closely related to the immune status of the organism [36]. Cancer cells can secrete immunosuppressive factors that easily cause immune dysfunction in the host and further lead to cell mediation disorders. Therefore, it is important to improve immunity to prevent and treat cancers. In this paper, because TSP displayed immunostimulatory activity, we hypothesized that it might also pose anticancer properties. Thus, we examined its activity on HepG2 cells. We demonstrated that TSP showed an inhibitory effect on HepG2 cells. The highest inhibitory rate was 66.8% at a concentration of 250 µg/mL, which was higher than that of *spirulina* polysaccharide [37] and *Pavlova viridis* polysaccharide [31]. Furthermore, the mechanism of TSP anticancer activity was mainly through inducing cell apoptosis rather than affecting the cell cycle and mitosis of HepG2 cells. These results were different from those of other studies [37,38,39], which report a mechanism of anticancer activity by blocking the G0/G1 phase of cancer cells. This result might be related to the differences in the chemical components and structure of the polysaccharides. This topic, along with the immunostimulatory and anticancer activities of TSP in vivo, requires further investigation. We will continue to study these topics.

## 4. Materials and Methods 

### 4.1. Tribonema sp. Samples and Reagents

*Tribonema* sp. was cultured in seawater BG11 solution. The culture conditions were in accordance with those of Wang et al. [7]. Simply, *Tribonema* sp. was cultured in a 40 L glass paneled tank containing 30 L of BG11 solution with 1.5% CO_2_ bubbling under 100 µmol photons m**^−^**^2^ s**^−^**^1^ of artificial light at 25 ± 2 °C for 7 days. Then, algae were collected by a 300 mesh silk screen and freeze-dried. The dried algal powder was kept at −20 °C before use. All the reagents used were of analytical grade and commercially available unless otherwise stated.

### 4.2. Extraction of Polysaccharides from Tribonema sp. (TSP)

The dried *Tribonema* sp. powder was extracted by the Soxhlet method with ethanol to remove pigments and lipids. The residue was then dried in an oven at 50 °C, and polysaccharides were extracted by hot distilled water (80 °C, 2 h) with the assistance of ultrasonic methods (20 times, 10 s working, 10 s rest, at 380 W power). Then, the mixture was filtered, and the solution was condensed by a rotary evaporator (RV10, IKA, German) and dialyzed for salt removal. The dialyzed solution was condensed again, and the final condensed solution was freeze-dried to obtain the purified sulfated polysaccharides called TSP.

### 4.3. Chemical Characterization

The M**_w_** of TSP was measured by HPLC with a TSK gel G5000PWxl column (Tosoh, Tokyo, Japan) using 0.05 mol/L Na_2_SO_4_ as the mobile phase on an Agilent 1260 HPLC system (Santa Clara, CA, USA) equipped with a refractive index detector (Santa Clara, CA, USA). The column temperature was 35 °C, and the flow rate of the mobile phase was 0.5 mL/min. Dextran standards (1 kDa, 5 kDa, 12 kDa, 50 kDa, 80 kDa, 270 kDa, and 670 kDa; Sigma, Mendota Heights, MN, USA) were used to calibrate the column.

The total sugar content of TSP was analyzed by the phenol-sulfuric acid method [40] using galactose as the standard. Sulfate content was determined by the barium chloride gelatin method [7]. The molar ratios of the monosaccharide composition were determined according to the method of Sun et al. [41]. 1-Phenyl-3-methyl-5-pyrazolone (PMP) precolumn derivation HPLC was used to determine the molar ratio of the monosaccharide composition. Mannose, rhamnose, fucose, galactose, xylose, glucose, and glucuronic acid from Sigma-Aldrich (St. Louis, MO, USA) were used as standards. FTIR spectra of TSP were determined on a Nicolet-360 FTIR spectrometer (Thermo, Waltham, MA, USA) between 400 cm**^−^**^1^ and 4000 cm**^−^**^1^.

### 4.4. Immunological Activity Determination

RAW264.7 cells (1 × 10^5^ cells/mL) were seeded in 12-well plates and incubated in an RPMI-1640 medium containing 10% fetal bovine serum. After 24 h, the medium was discarded and the macrophages were washed with PBS. Then, TSP was added to the cells at various concentrations (12.5, 50, and 200 µg/mL). After 24 h of treatment, some of the cells were used to determine macrophage cell viability according to the method of Sung et al. [42]. Some cells were centrifuged at 1000 rpm for 10 min at 4 °C, and the supernatants were collected and stored at −80 °C. Cytokines, such as IL-6, IL-10, and TNF-α, of RAW264.7 macrophages were determined by enzyme-linked immunosorbent assay (ELISA) kits (Lengton, Shanghai, China). LPS (1 µg/mL) and PBS were used as the positive and negative controls, respectively.

### 4.5. Anticancer Activity of TSP

#### 4.5.1. Cell Culture 

HepG2 cells (supplied by Kunming Cell Bank, Chinese Academy of Sciences, Kunming, China) were cultured in DMEM containing 10% fetal bovine serum, 100 U/mL penicillin, and 100 mg/mL streptomycin at 37 °C with 5% CO_2_. 

#### 4.5.2. Evaluation of Inhibiting HepG2 Growth Activity In Vitro

The inhibitory activity of TSP on cell growth at various concentrations (50, 100, 150, 200, and 250 µg/mL) was assessed by the MTT assay. The cells were seeded in a 96-well plate at a concentration of 1 × 10^4^ cells/mL and incubated for 48 h with various concentrations of TSP. Then, 200 µL of 0.5 mg/mL MTT solution was added to each well. After 4 h of incubation, the plates were centrifuged for 10 min at 8000 rpm. The MTT solution was removed, and then 200 µL of DMSO was added to each well. The absorbance at 570 nm was determined.

#### 4.5.3. Apoptosis Assessment

The apoptosis states of HepG2 cells under treatment with TSP were determined by an Annexin V-FITC/PI apoptosis kit (Jiancheng, Nanjing, China). The cells were collected and washed twice with ice-cold PBS. Then, the cells were diluted to 1 × 10^6^ cell/mL with binding buffer. The diluted cells were dyed with 10 μL of Annexin V-FITC for 30 min at room temperature and stained with 5 μL of propidium iodide (PI) for 5 min. After incubation, the apoptosis of cells was determined by flow cytometry.

#### 4.5.4. Analysis of the Cell Cycle

A cell cycle analysis kit (Beyotime, Shanghai, China) was used to analyze the cell cycle. The cells were plated in DMEM with different concentrations of TSP for 24 h. Then, the mixture was placed into the flow cytometry tube and centrifuged at 1500 rpm for 5 min to obtain cell pellets. After that, the cell pellets were washed with precooled PBS and fixed in ice-cold 70% ethanol overnight at 4 °C. Fixed cells were rewashed with PBS and incubated with propidium iodide (PI) staining solution (0.5 mL of staining buffer, 25 μL of PI staining solution, and 10 μL of RNAase A) for 30 min at 37 °C in the dark. Cell cycle analysis was conducted with Guava^®^ easyCyte 6-2L (Millipore, Billerica, MA, USA) using 10,000 counts per sample. The percentage of cells distributed in the different phases of G0/G1, S, and G2/M were recorded and analyzed. 

### 4.6. Statistical Analysis

All data are shown as the means ± SD (standard deviation) of three repeated experiments to ensure the reproducibility of the results. Statistical analysis was performed using SPSS (17.0 for windows, IBM, Hong Kong, China). The difference among groups was analyzed by one-way ANOVA.

## 5. Conclusions

In this paper, we extracted TSP from *Tribonema* sp., partially characterized its chemical composition and structure, and determined its immunostimulation and anticancer activities on macrophage cells and HepG2 cells. The results showed that TSP is a sulfated heteropolysaccharide with a M**_w_** of 197 kDa and is mainly composed of galactose. TSP showed significant immune-modulatory activity by enhancing macrophage cell viability and upregulating some cytokines, such as IL-6, IL-10, and TNF-α. It also showed significant dose-dependent anticancer activity on HepG2 cells. The cycle analysis indicated that the mechanism of TSP anticancer activity is mainly the induction of cell apoptosis rather than affecting the cell cycle and mitosis of HepG2 cells. These findings indicated that TSP has potential as an alternative to treat cancers. The exploitation of new biological activities of polysaccharides from microalgae will improve the public acquaintance and widen the application of microalgae, except the biorefinery area. However, a long process is needed to confirm its effect, including in vivo experiments to define its toxicology and function mechanisms.

## Figures and Tables

**Figure 1 molecules-24-00322-f001:**
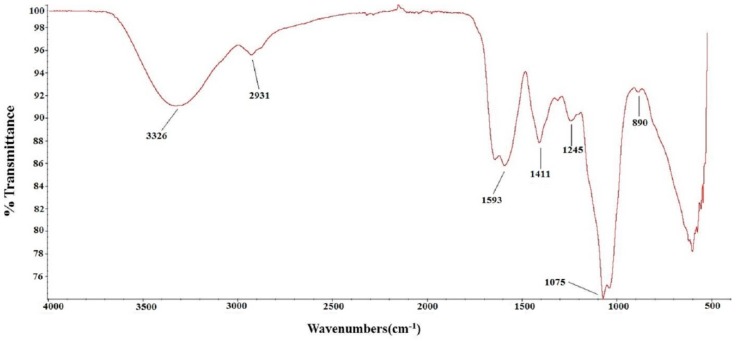
FT-IR of TSP.

**Figure 2 molecules-24-00322-f002:**
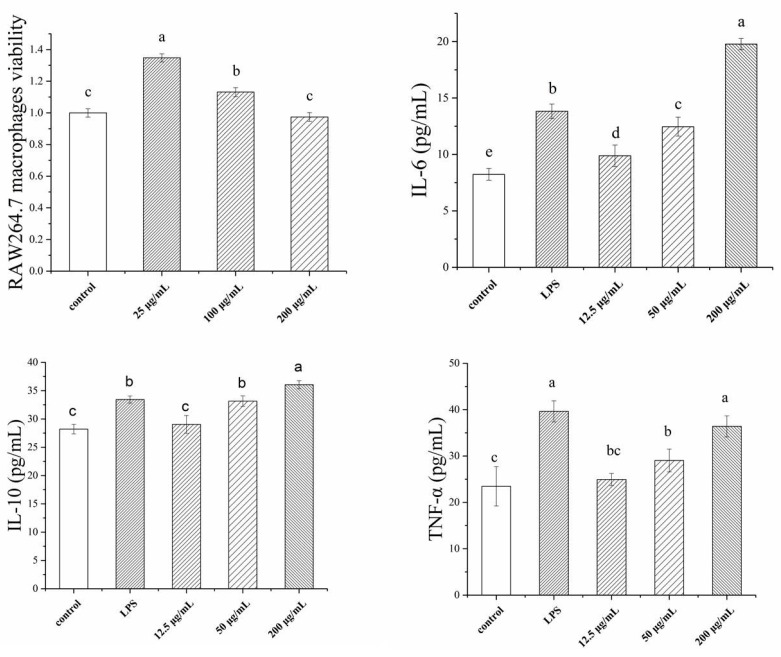
Immunological activity of RAW264.7 macrophages, including macrophages viability, IL-6, IL-10, and TNF-α index, with different concentrations of TSP (a, b, c, d denote significant difference at *p* < 0.05).

**Figure 3 molecules-24-00322-f003:**
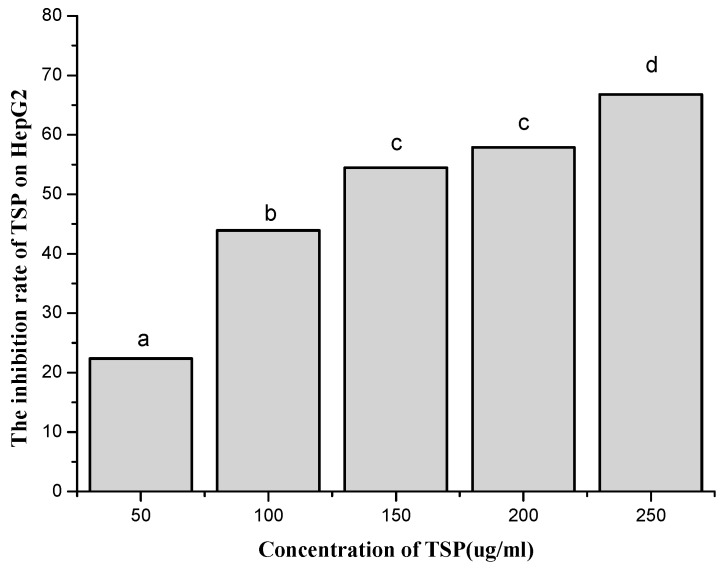
The effect of different concentrations of TSP on the inhibition rate of HepG2 by MTT assay (a, b, c, d denote significant difference at *p* < 0.05).

**Figure 4 molecules-24-00322-f004:**
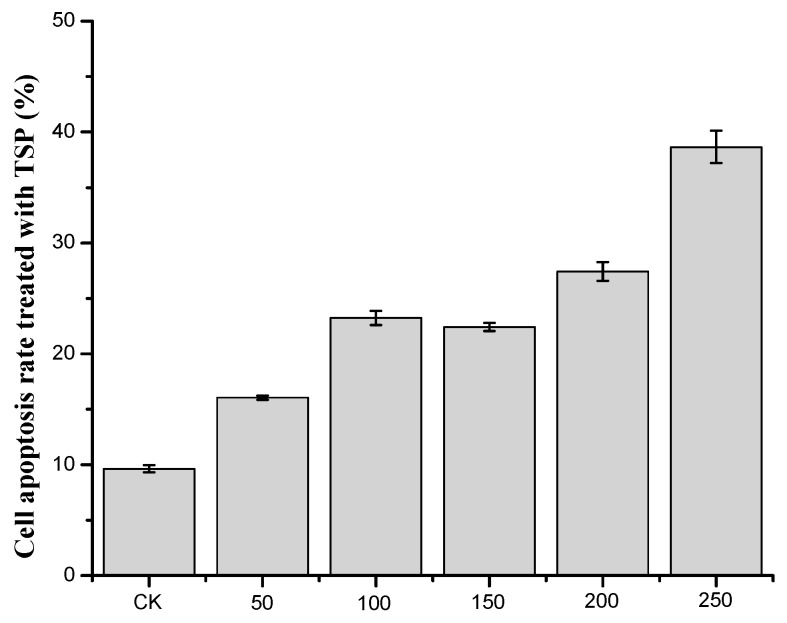
The apoptosis rate of cells treated with different concentrations of TSP.

**Figure 5 molecules-24-00322-f005:**
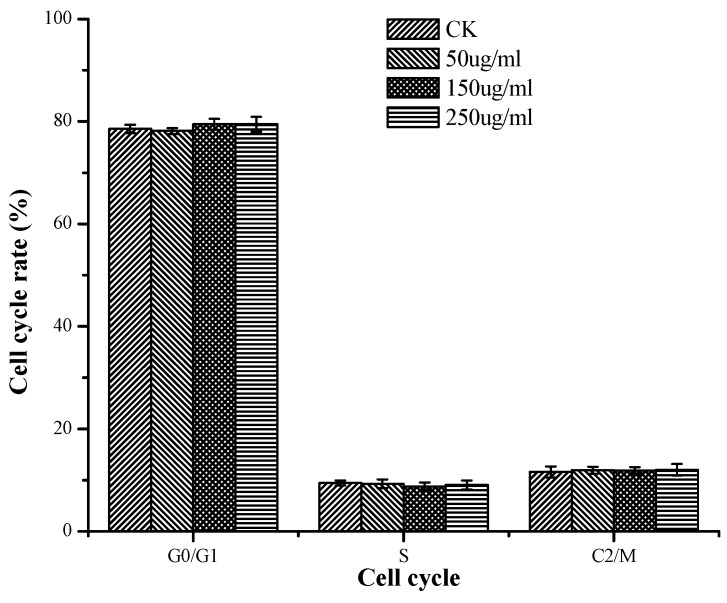
The cell cycle rate of cells treated with different concentrations of TSP.

**Table 1 molecules-24-00322-t001:** Chemical composition of TSP (% *w*/*w* dry weight for total sugar and sulfate, molar ratio for monosaccharides composition).

Sample	Total Sugar/%	Sulfate/%	M_w_/kDa	Monosaccharides Composition (Molar Ratio)						
				Man	Rha	GlcA	Glc	Gal	Xyl	Fuc
TSP	32.5 ± 0.2	21.9 ± 0.15	197 ± 0.6	0.00 ± 0.00	0.101 ± 0.002	0.063 ± 0.001	0.069 ± 0.004	1.00 ± 0.03	0.32 ± 0.01	0.095 ± 0.005

Man: mannose; Rha: rhamnose; Glc A: glucuronic acid; Gal: galactose; Glc: glucose; Xyl: xylose; Fuc: fucose.

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
