# Peer review of "Partial Characterization, the Immune Modulation and Anticancer Activities of Sulfated Polysaccharides from Filamentous Microalgae Tribonema sp."

_molecules, 2019, doi:10.3390/molecules24020322_

Round 1
Reviewer 1 Report
The author needs to focus the introduction on the activity rather than splitting it with the biofuels. The author needs to clarify details of table 1 and since the MW of TSP is not that high, why the author did not make use of more reliable tools to validate the structure and physical characteristics of TSP? Rather than the FTIR and monosaccharide composition which are not providing solid answers. IR can even be moved to SI. Figure caption needs to contain identifications of all the symbols. The author needs to be more specific in providing the information such as “According to our knowledge, there is no study reporting on polysaccharides from Tribonema.” The author should highlight the advantages and disadvantages of TSP as possible drug therapy. The manuscript needs to be proofread for few typos and formatting issues. Supporting information needed to be added.
Author Response
The author needs to focus the introduction on the activity rather than splitting it with the biofuels. The author needs to clarify details of table 1 and since the MW of TSP is not that high, why the author did not make use of more reliable tools to validate the structure and physical characteristics of TSP? Rather than the FTIR and monosaccharide composition which are not providing solid answers. IR can even be moved to SI. Figure caption needs to contain identifications of all the symbols. The author needs to be more specific in providing the information such as “According to our knowledge, there is no study reporting on polysaccharides from Tribonema.” The author should highlight the advantages and disadvantages of TSP as possible drug therapy. The manuscript needs to be proofread for few typos and formatting issues. Supporting information needed to be added.
Response: Because TSP was not a purified sample, it was difficult to characterize its structure by other tools such as NMR. In my opinion, we should firstly prove TSP had immunostimulation and anticancer activities. Then, we could purify TSP and further characterize the structure. And determine the effect in vivo, which will be our next goal. If possible, IR can be supplied as SI. Figure captions have been revised and included all the symbols. The sentence “According to our knowledge, there is no study reporting on polysaccharides from Tribonema.” has been revised. We have high lightened the advantages and disadvantages of TSP as possible drug therapy in Conclusion. The manuscript has been reviewed and revised.
Reviewer 2 Report
Chen and co-authors proposed a study reporting the characterization of sulfated polysaccharides from Tribonema sp associated to biological activities (immune-modulation and anticancer). The experimental work is, in general, well done. The statements are sometimes too strong, sometimes missing. I give a positive opinion on this work, but it needs some corrections. The authors should address the following points. The authors should also cite the proposed papers to enhance the background but also other appropriated references before any final acceptance. Besides, the manuscript is well written but should be revised a last time by a native English speaker.
Please, find below few comments (not all) ...
For the title, the statement about characterization is too strong regarding the real characterization done. At least, please change it to Partial characterization.
Concerning the Fig 1, the authors should improve its quality and increase typo (hard to read the captions for each axis, same for labeling).
Table 1: A lot of spaces are missing (title for example between the bracket and %. Another one for Xyl: xylose; etc…)… Values of deviation must be added. Each monosaccharide should begin with a Cap, e.g. Glc: Glucose. Please also change Glc A to GlcA.Is it really a molar ratio or a ratio calculated with Gal as reference 1… Please check again the values, especially if the main title claims: %w/w dry weight (which is not compatible with a molar ratio of monosaccharide) …
Few mistakes are still in the document and should be reviewed. Please read the guide for authors for proper referencing all through the manuscript.
Spaces are missing throughout the manuscript next to each bracket citation.
L.27: MTT abbreviation should be detailed for the first time. Same comment for Interleukin.
L.28: Change 66.78% to 66.8%
L.30: in vivo should be in italic. Please check each Latin term all over the paper for amendments.
L.36: Few lines should be added to enhance the introduction about the concept of bioraffinery and the need to valorize primary and secondary metabolites for a real future in biofuel industry (niche market for polysaccharides, as example). The authors should cite this paper here (https://doi.org/10.1016/j.biotechadv.2016.08.001).
L.40-46: The authors must discuss (and cite) about previous results detailed in the literature concerning Tribonema and its polysaccharides. A paragraph must be added to enhance the substance of the introduction (which is light).
L.47: Please cite also here, since Porphyridium and Arthospira are cited as example (https://doi.org/10.1016/j.biotechadv.2018.11.014; https://doi.org/10.1016/j.algal.2015.04.014).
L.50 Tribonema should be in italic
L.61-62: The statements are too strong here, especially for finding new anticancer drugs. Please change the sentence. The authors should explain here in few lines the main goals and strategy of the paper (3-4 lines max).
L.65: Few lines should be added for comparing this very low yield to the literature. Few lines should be also added to discuss the result (and the extraction/purification procedure which is probably responsible of this %w ratio).
L.72: For Mw, w should be in subscript. Any information of Mn? About the polydispersity Index (PDI)? If possible, add the values. PDI is particularly important especially for papers focusing on biological activities (o be sure we are talking about one class of monodisperse polysaccharide). By the way, please compare this value to the data you can find in the literature (in the discussion section).
Author Response
Chen and co-authors proposed a study reporting the characterization of sulfated polysaccharides from Tribonema sp associated to biological activities (immune-modulation and anticancer). The experimental work is, in general, well done. The statements are sometimes too strong, sometimes missing. I give a positive opinion on this work, but it needs some corrections. The authors should address the following points. The authors should also cite the proposed papers to enhance the background but also other appropriated references before any final acceptance. Besides, the manuscript is well written but should be revised a last time by a native English speaker.
Response: We have revised the English by American Journal Experts.
Please, find below few comments (not all) ...
1. For the title, the statement about characterization is too strong regarding the real characterization done. At least, please change it to Partial characterization.
Response: We have revised the title according to the comments.
2. Concerning the Fig 1, the authors should improve its quality and increase typo (hard to read the captions for each axis, same for labeling).
Response: We have revised Fig.1 and improved the quality.
3. Table 1: A lot of spaces are missing (title for example between the bracket and %. Another one for Xyl: xylose; etc…)… Values of deviation must be added. Each monosaccharide should begin with a Cap, e.g. Glc: Glucose. Please also change Glc A to GlcA.Is it really a molar ratio or a ratio calculated with Gal as reference 1… Please check again the values, especially if the main title claims: %w/w dry weight (which is not compatible with a molar ratio of monosaccharide) …
Response: Table 1 has been revised. In title, %w/w dry weight was for total sugar and sulfate results and we have revised the title.
4. Few mistakes are still in the document and should be reviewed. Please read the guide for authors for proper referencing all through the manuscript.
Response: We have reviewed the manuscript and revised it.
5. Spaces are missing throughout the manuscript next to each bracket citation.
Response: We have added the spaces.
6. L.27: MTT abbreviation should be detailed for the first time. Same comment for Interleukin.
Response: We have added the details of MTT, Interleukins and TNF-α.
7. L.28: Change 66.78% to 66.8%
Response: 66.78% in this paper has been revised to 66.8%.
8. L.30: in vivo should be in italic. Please check each Latin term all over the paper for amendments.
Response: In vivo has been written in italic. And we have checked all the Latin terms.
9. L.36: Few lines should be added to enhance the introduction about the concept of bioraffinery and the need to valorize primary and secondary metabolites for a real future in biofuel industry (niche market for polysaccharides, as example). The authors should cite this paper here (https://doi.org/10.1016/j.biotechadv.2016.08.001).
Response: We have added some sentences to introduce the concept of bioraffinery and cited the paper.
10. L.40-46: The authors must discuss (and cite) about previous results detailed in the literature concerning Tribonema and its polysaccharides. A paragraph must be added to enhance the substance of the introduction (which is light).
Response: I am sorry that I did not retrieve the relative reference of polysaccharide of Tribonem. And we have revised the expression.
11. L.47: Please cite also here, since Porphyridium and Arthospira are cited as example (https://doi.org/10.1016/j.biotechadv.2018.11.014; https://doi.org/10.1016/j.algal.2015.04.014).
Response: We have added the paper.
12. L.50 Tribonema should be in italic
Response: Tribonema has been written in italic.
13. L.61-62: The statements are too strong here, especially for finding new anticancer drugs. Please change the sentence. The authors should explain here in few lines the main goals and strategy of the paper (3-4 lines max).
Response: We have revised the sentence and added the mail goals of the paper.
14. L.65: Few lines should be added for comparing this very low yield to the literature. Few lines should be also added to discuss the result (and the extraction/purification procedure which is probably responsible of this %w ratio).
Response: We have added some sentence to compare the yields to the literature.
15. L.72: For Mw, w should be in subscript. Any information of Mn? About the polydispersity Index (PDI)? If possible, add the values. PDI is particularly important especially for papers focusing on biological activities (o be sure we are talking about one class of monodisperse polysaccharide). By the way, please compare this value to the data you can find in the literature (in the discussion section).
Response: “w” in Mw has been written in subscript. We have not determined the Mn and PDI. In my opinion, we should firstly prove TSP had immunostimulation and anticancer activities. Then, we could purify TSP and further characterize the structure including Mn, PDI etc. and determine the effect in vivo, which will be our next goal.